# Smart Flexible 3D Sensor for Monitoring Orthodontics Forces: Prototype Design and Proof of Principle Experiment

**DOI:** 10.3390/bioengineering9100570

**Published:** 2022-10-18

**Authors:** Soobum Lee, Chabum Lee, Jose A. Bosio, Mary Anne S. Melo

**Affiliations:** 1Department of Mechanical Engineering, University of Maryland, Baltimore County, 1000 Hilltop Circle, Baltimore, MD 21250, USA; 2J. Mike Walker’ 66 Department of Mechanical Engineering, Texas A&M University, 3123 TAMU, College Station, TX 77843, USA; 3Division of Orthodontics, Department of Orthodontics & Pediatric Dentistry, University of Maryland School of Dentistry, Baltimore, MD 21201, USA; 4Division of Operative Dentistry, Department of General Dentistry, University of Maryland School of Dentistry, Baltimore, MD 21201, USA

**Keywords:** orthodontics, dental care, technology, dental, orthodontic appliance design, dentistry

## Abstract

There is a critical need for an accurate device for orthodontists to know the magnitude of forces exerted on the tooth by the orthodontic brackets. Here, we propose a new orthodontic force measurement principle to detect the deformation of the elastic semi-sphere sensor. Specifically, we aimed to detail technical issues and the feasibility of the sensor performance attached to the inner surface of the orthodontic aligner or on the tooth surface. Accurate force tracking is important for the optimal decision of aligner replacement and cost reduction. A finite element (FE) model of the semi-sphere sensor was developed, and the relationship between the force and the contact area change was investigated. The prototype was manufactured, and the force detection performance was experimentally verified. In the experiment, the semi-sphere sensor was manufactured using thermoplastic polymer, and a high-precision mold sized 3 mm in diameter. The change in the contact area in the semi-sphere sensor was captured using a portable microscope. Further development is justified, and future implementation of the proposed sensor would be an array of multiple semi-sphere sensors in different locations for directional orthodontic force detection.

## 1. Introduction

Orthodontic treatment with clear aligners has become an increasingly popular method to correct dental malocclusions [1]. Clear aligner treatment (CAT) uses a transparent thermoplastic material to align the teeth, offering comfort and aesthetics [2]. Over the last two decades, CAT has evolved with digital planning, enhanced thermoplastic materials, and resin attachments bonded on teeth to enhance tooth movement [3]. CAT has also been applied in treating severe crowding and complex malocclusions [4]. Currently, over 10 million patients have been treated with clear aligner technology worldwide. In the United States, the number of patients treated with CAT has doubled within 6–7 years [5]. The clear aligners market was valued USD 2.71 billion in 2021 and is expected to reach USD 9.3 billion by 2030 [6]. With the increased demand for this treatment modality, concerns exist about root resorption regarding the forces employed to attain tooth movement [7,8].

In attempting to assess the orthodontic forces applied against the teeth quantitatively, many studies have assessed the magnitude and direction of forces during orthodontic treatment [9,10,11]. Fourteen transducers were used to create a laboratory-based human mouth model, and the error in force measurements was about 1.54% [12]. A design and fabrication of a simulated oral model for 3D orthodontic force measurements were created in the laboratory and illustrated that bi-component silicones with 2:8 ratios had similar mechanical properties to periodontal ligament (PDL), which could, in turn, represent a laboratory model resembling periodontal ligament (PDL) where force measurement could be measured [13]. Several investigations have sought advanced methodologies to monitor orthodontic forces over the years [5,14,15,16]. Most of them included resin jaw replicas and strain gauge-based systems and were designed as large, complex devices that are not user- or dentist-friendly and do not allow for real-time measurement. Recently, a refractive index-based surface plasmon resonance biosensor was investigated for caries diagnosis [17]. However, a comprehensive and up-to-date literature search still shows the lack of available evidence.

This work presents a low-cost, facile, smart, flexible 3D sensor endowed with monitoring protocol within a clear aligner. Here, we report the results of preliminary findings for prototype design and proof of concept of a new orthodontic force sensor applied to a clear aligner intended to establish analytical relationships between externally applied force and the sensor response.

## 2. Materials and Methods

Principle and Method Analysis

Figure 1 displays the sensor setup over the clear aligner attachment. Based on the clear aligner technique, attachments made of dental composites are bonded over the tooth’s surface. An attachment’s most basic function is retention and force guidance. In order to fully realize the intended tooth movements, the aligners have to be fully seated and embrace the composite attachment. Aligners have the flexibility that needs to be compensated when seated onto dental units that require movement. The second function of attachments is to facilitate movement by changing the tooth’s surface area to provide an active surface for pushing (Figure 1).

The active surface of the optimized attachment delivers the desired forces while simultaneously extruding, rotating, or providing root control. Each attachment has a specific surface profile on which the aligner provides pressure (Figure 1). The sensor system aims to determine whether the force on the surface of the attachment is symmetrical and uniform and quantify the applied force over the tooth, thereby ensuring the quality of orthodontic movement.

The sensor is made of biocompatible polymer (polydimethylsiloxane (PDMS) or dimethicone). The PDMS is a polymer widely used to fabricate and prototype microfluidic chips [16]. This material is cured in a semi-sphere geometry in the ultra-precision mold (stainless steel, diameter = 3 mm). The detailed material properties of the thermoplastic polymer are defined in Table 1. The change in the orthodontic force can be detected by changes to the contact area between the sensor and the aligner’s inner surface. A portable microscope was used to detect the area change. (Figure 2). The sensor dimension is for demonstration purposes and can be reduced as long as the microscope resolution allows for the detection of its area change. This simple concept can operate more reliably compared to other flexible sensors and avoid complex implementation of resistive sensors or electronics on a small scale.

### 2.1. Finite Element Analysis of the Sensor

As the first step of this study, the finite element model for the aligner force sensor was built. Conceptually, a semi-sphere sensor is bonded to an attachment or even to a tooth, and the sensor is in contact with the inner surface of the aligner. The finite element modeling is made for one semi-sphere sensor and a contact part of aligner material, as shown in Figure 3. A quarter model is considered for simplicity and analysis efficiency, and the tooth (enamel) part is removed. A small force applied to the sensor is acceptable to assign linearly isotropic properties for each material (sensor and aligner).

ANSYS workbench 2021 is used for FEM analysis of the model. In this study, the authors were interested in understanding the force-area relationship by a normal direction force. The aligner parts are simplified as a solid rectangular block (1.5 × 1.5 × 1.5 mm^3^) in contact with a quarter of the semi-spherical sensor (radius = 1.5 mm). (Figure 4) To obtain a reliable solution from nonlinear contact analysis, the model is refined with a sufficiently small element size: 0.04 mm close to the contact area and 0.4 mm elsewhere. In total, 29,525 nodes and 17,396 elements are used: 16,237 nodes and 9983 elements for the block; and 13,288 nodes and 7413 elements for the semi-sphere. The semi-sphere makes frictional contact with the resin attachment. The bottom surface of the lower plate was set as fixed, and an incremental vertical displacement was applied (step size = 0.025 mm, up to 0.30 mm) to the top of the sphere surface, while the corresponding symmetric boundary (sliding) conditions are applied. Changes in the vertical reaction force and the contact area were traced (Figure 4).

### 2.2. Prototype of the Force Detection Sensor

The sensor design proposed in the previous section was carefully prototyped for experimental verification. In the clear aligner system, the proposed semi-sphere sensor placement is considered. Several factors may affect the force reading of the sensor, such as the type and shape of the sensor material and the position of the sensor. In Figure 1, a possible placement of the sensor on the attachment or tooth was delineated. Several advantages of placing the sensor on the attachment can be observed. First, the attachment is similar to an anchor for the aligner, and a more significant force concentration is expected to measure a larger orthodontic force more accurately. Second, the sensor can be attached to any of the sides of the attachment, thus producing a desired 3D force measurement.

### 2.3. Experimental Force Evaluation

As a first trial, a single semi-sphere sensor was fabricated. The silicon material was cured in the precision mold, and after several trials, a good surface quality was obtained with no bubbles inside. A commercial orthodontic cement Transbond XT (3 M Unitek, Monrovia, CA) is consisted of silane-treated quartz (70–80% by weight), bisphenol-A-diglycidyl ether dimethacrylate (10–20%), bisphenol-A-bis (2-hydroxyethyl) dimethacrylate (5–10%), silane-treated silica (<2%) and diphenyliodonium hexafluorophosphate (<0.2%). Cement to bond the semi-sphere lens on the flat plate that imitates the tooth surface was used, and another transparent flat plate pressures the top of the sensor (ProForm .030” (1 mm) Splint Material) that imitates the clear aligner. The contact area change between the sensor and the top plate was captured by a portable digital microscope (STPCTOU, Model PF018005), as shown in Figure 5.

## 3. Results & Discussion

For the accurate image capture of the dynamic contact area, the force vector tracking software was implemented using Matlab that reads the movie clip from the microscope and tracks the center and contact surface area. The software run example is shown in Figure 6, where the contact area change is calculated by image processing (into black and white) and counting white pixels. The optimum forces for orthodontic tooth movement range from 10~20 g (intrusion) to 0.1~0.2 N (intrusion) to 0.7~1.2 N (bodily translation), depending on the type of movement. For the proof of concept, an excessive force that varies by the amplitude up to 7.0 N was applied.

Figure 3 displays the force vs. contact area graph from Finite Element analysis (dashed lines) and test (* marks). The result shows an increasing trend of the area by the increased force. The rate of the contact area to the force (slope) decreases as the force increases, indicating that it becomes harder to make the same change in area as the force is higher. The silicon PDMS sensor’s material properties might vary depending on the curing process. Thus, the area measured by the change in Young’s modulus (±8%) that covers most of the testing points was investigated. This simulation causes the variance of force evaluation from the same contact area. For example, when Young’s modulus is 17.0 MPa (nominal value), the contact force is 1 N by the contact area is 0.51 × 10^−6^ m^2^. When Young’s modulus variation is assumed by ±8%, the force measurement changes from 0.90 N~1.11 N (−10%~+11%). This variance increases when the force increases as well: −0.27~+0.12 N (−13.5%~6.0%) from 2 N, −0.22~+0.29 N (−7.3%~+9.6%) from 3 N, and −0.55~0.27 N (−13.7%~+6.7%) from 4 N, respectively. In summary, we observed approximately ±10% variation of force evaluation when there is the uncertainty of material property by ±8% due to errors caused by semi-spheres fabrication process (raw material properties, heating/curing process, and spheres dimension variance), and it covers the range of force from the experimental study. The nominal FEM data (17.0 MPa) follows a second order polynomial, Area (m^2^) = −1.19 × 10^−6^ f2 + 2.95 × 10^−7^ f + 1.98 × 10^−7^ (f = force in N), and the error to each measured data is also shown in Table 2 (average = 6.54%, standard deviation = 13.30%).

As a proof-of-concept study, we presented the earliest point in the development process that presents a reasonable likelihood of success. The current preliminary work is limited in terms of the scale and direction of the force for successful commercialization. Further investigation should look at different load situations and compare them to other measurement technology documented in the literature. Another aspect investigated was tooth movement relative to the alveolar bone and, consequently, the biological processes that would facilitate movement corresponding to these measured forces. This can be achieved by subsequent phases of investigation where more complex finite elements consider tooth and periodontal ligament and clinical investigation where the biological component could be considered. Moreover, we need to identify unreliable deformation and related strains/moments of the attachments and compensate them.

The key objective of this research was to detail technical parameters to guide the design assessed by visual inspection of results. This objective is rooted in the desire to develop a new sensor to detect force applied to dental units and better control teeth movements using thermoplastic clear aligners.

The study revealed that the sensor deformation reading could demonstrate the force applied to specific sensor parts. However, only unidirectional forces could be detected. This preliminary pilot study showed the ability to better understand the amount and direction of forces created by aligners on the teeth. If color-coded detection of forces applied to the sensor can be found, then computer-generated “Clinchecks” will be able to demonstrate in real-time if forces are more significant than the periodontal tissue can absorb. The results in Figure 6 are a sample of the extracted visual observations.

## 4. Conclusions

This proof-of-concept study supports the application of a flexible semi-sphere sensor for orthodontic forces monitoring. The preliminary data of the sensor revealed its ability to detect forces applied to the aligner attachment. Efforts are underway to build on this preliminary work to develop a flexible-based biosensor for continuous non-invasive force monitoring. While key challenges remain toward such a long operation, this preliminary proof-of-concept demonstration indicates the potential of the sensor. However, intra-oral evaluations of the flexible semi-sphere sensor are further needed to demonstrate the ability to detect the multidirectional forces applied to the tooth. Future efforts aim to address these challenges, integrate the corresponding electronic backbone for powering the sensor, signal processing, and wireless communication on a flexible wearable platform, and perform a large-scale orthodontic monitoring study. We will also investigate the sensor responses by unnormal force, possibly measured from the area change and sliding distance. Finally, a functional interface (handheld microscope device, database of captured images, and the force tracking record) will be devised to make the developed concept clinically applicable.

## Figures and Tables

**Figure 1 bioengineering-09-00570-f001:**
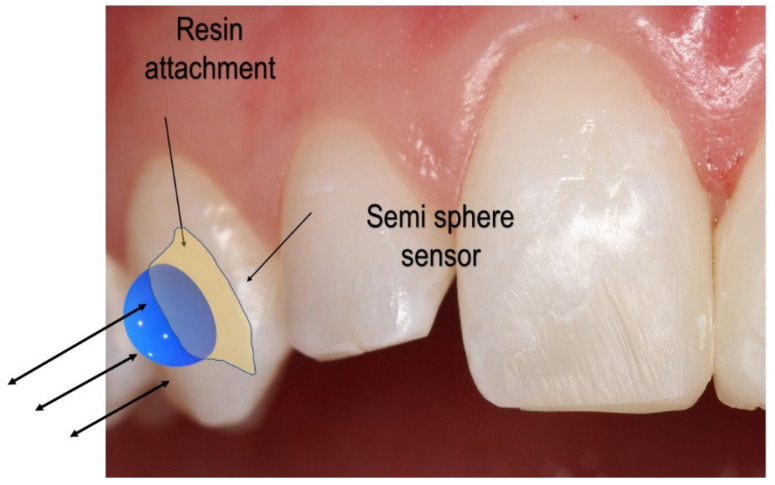
Proposed sensor placement over the attachment.

**Figure 2 bioengineering-09-00570-f002:**
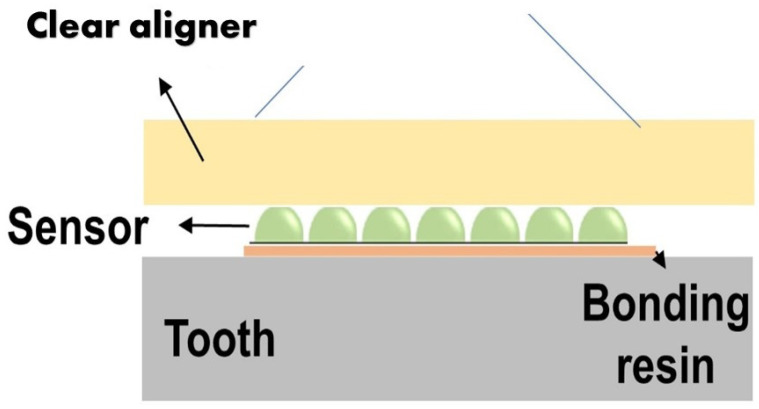
Proposed sensor system architecture.

**Figure 3 bioengineering-09-00570-f003:**
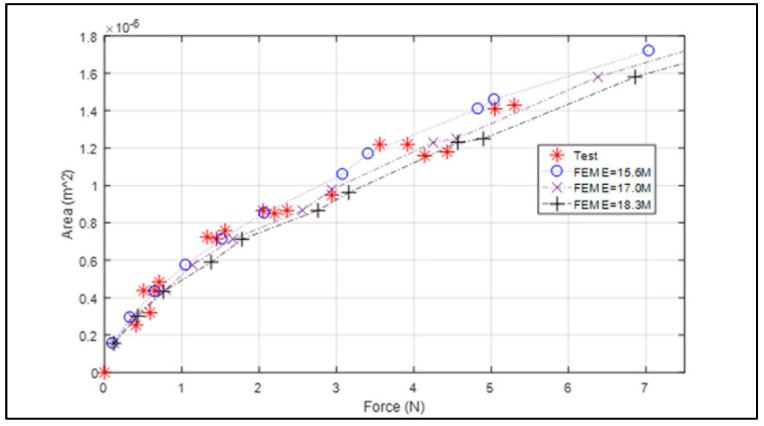
Contact area vs. force by FEM and experiment.

**Figure 4 bioengineering-09-00570-f004:**
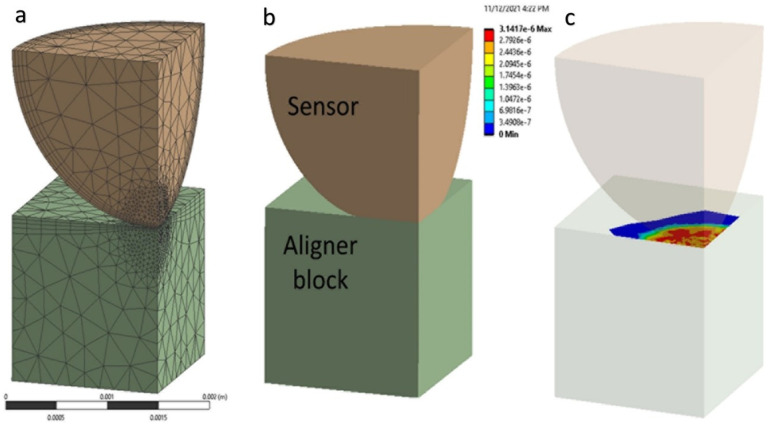
FEM contact simulation for the orthodontic sensor: (**a**) FEM mesh (**b**) deformed geometry (**c**) surface pressure.

**Figure 5 bioengineering-09-00570-f005:**
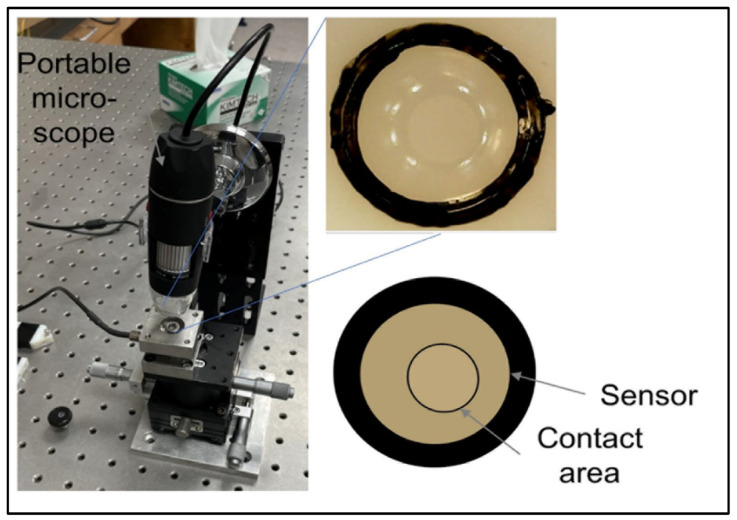
Portable microscope and sensor deformation area concept.

**Figure 6 bioengineering-09-00570-f006:**
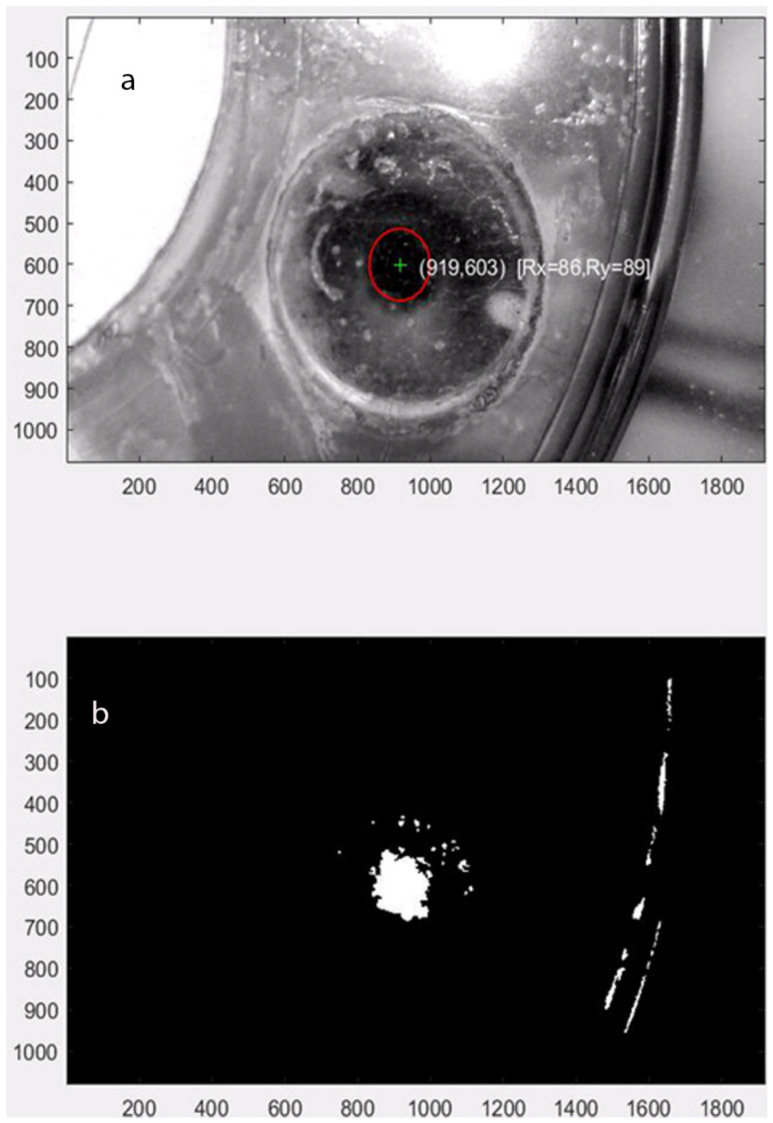
The force measurement setup using prototype: (**a**) force measurement setup (**b**) image processing (black and white) to calculate contact area.

**Table 1 bioengineering-09-00570-t001:** Material properties of the finite element model for the aligner sensor.

Material	Young’s Modulus	Poisson’s Ratio
Tooth—enamel	80 GPa	0.3
Sensor—silicon PDMS	17 MPa	0.43
Aligner	1.5 GPa	0.43

**Table 2 bioengineering-09-00570-t002:** Force calibration.

F [N]	0.408	0.507	0.590	0.650	0.709	1.326	1.454	1.560	2.047	2.199
S [mm_2_]	0.253	0.438	0.320	0.438	0.484	0.725	0.715	0.759	0.866	0.851
FEM [mm^2^]	0.032	0.035	0.037	0.039	0.040	0.057	0.060	0.063	0.075	0.079
Error [%]	−20.2	27.15	−13.18	13.71	20.5	27.58	18.83	20.55	15.13	7.84
F [N]	2.359	2.939	3.556	3.918	4.14	4.432	5.049	5.301	8.224	
S [mm_2_]	0.867	0.949	1.216	1.222	1.157	1.180	1.411	1.431	1.780	
FEM [mm^2^]	0.083	0.096	0.110	0.117	0.122	0.127	0.139	0.143	0.182	
Error [%]	4.68	−1.43	10.8	4.33	−4.81	−7.23	1.91	0.19	−2.21	

## Data Availability

Data is contained within the article.

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
