# Peer review of "Smart Flexible 3D Sensor for Monitoring Orthodontics Forces: Prototype Design and Proof of Principle Experiment"

_bioengineering, 2022, doi:10.3390/bioengineering9100570_

Round 1
Reviewer 1 Report
The author proposed a kind of elastic semi-sphere sensors to measure orthodontic forces through deformation of semi-sphere shapes. The measurement principle, sensor fabrication, and finite element calculations were carried out. The manuscript is publishable after minor revisions.
1. Some sentences are repeated. Such as 'the relationship between the force and the contact area was investigated' on Line 22 and Line 26 in Abstract.
2. Some words should be re-spelled, such as "USD" on Line 43 and Line 44 in Introduction should be "US dollar" or "$".
3. Line 53, 'PDL' is not defined.
4. There is no Figure 1A (Line 74) and Figure 1B (Line 79).
5. Line 87, reference '21' should be [16].
6. What's the measurement errors of the force in Figure 3? The force is calculated from the contact area change from optical microscopy. The measurement error of contact areas would affect the accuracy of measured forces. Compared with the present techniques and clinic requirements, is the accuracy of the designed sensors good enough for practice applications?
7. If the force is not vertical to the semi-spherical sensors, but inclined, how about the measured forces?
8. Line 149 - Line 152: gram is the unit of mass, not unit of force. Please use the units of force.
9. Some typos, such as Line 164, "0.51E-6 m2" should be '0.51 X 10-6 m2' or '0.51 E-6 m2'.
Author Response
Dear Reviewer,
First, the authors would like to thank the time spent reviewing our manuscript entitled " Smart Flexible 3D sensor for monitoring orthodontics forces: Prototype Design and Proof of Principle experiment." We believe that all the considerations improved the quality of the manuscript, and we reiterate our wish to publish at Polymers. The answers to your comments are below point-by-point. The document was reviewed. The modifications were written in red in the revised manuscript.
REVIEWERS COMMENTS:
REVIEWER #1
- Some sentences are repeated. Such as 'the relationship between the force and the contact area was investigated' on Line 22 and Line 26 in Abstract.
A: Thank you. The repeated sentences were deleted.
- Some words should be re-spelled, such as "USD" on Line 43 and Line 44 in Introductionshould be "US dollar" or "$".
A: Thank you. The sentence was corrected in the revised manuscript.
- Line 53, 'PDL' is not defined.
A: Thank you. The sentence was corrected in the revised manuscript.
- There is no Figure 1A (Line 74) and Figure 1B (Line 79).
A: Thank you. The sentence was corrected in the revised manuscript.
- Line 87, reference '21' should be [16].
A: Thank you. The reference was corrected in the revised manuscript.
- What's the measurement errors of the force in Figure 3? The force is calculated from the contact area change from optical microscopy. The measurement error of contact areas would affect the accuracy of measured forces. Compared with the present techniques and clinic requirements, is the accuracy of the designed sensors good enough for practice applications?
A: Thank you for your comment. Table 2 is updated with error measures to indicate the difference from the FEM simulation. The average error is about 6%, with a standard deviation of about 13%. We understand this precision may not be sufficient for clinical needs, but as an early phase feasibility study, we believe it is a successful implementation using an easy concept of force measurement with plenty of improvement remain.
- If the force is not vertical to the semi-spherical sensors, but inclined, how about the measured forces?
A: The current sensor concept is valid for measuring the force's normal (or vertical) direction. Sensor design update for measuring an inclined force remains as future work. The following sentence is added in Conclusion: “We will also investigate the sensor responses by unnormal force, that can be possibly measured from the area change as well as sliding distance.”
- Line 149 - Line 152: gram is the unit of mass, not unit of force. Please use the units of force.
A: Thank you. The sentence was corrected in the revised manuscript.
- Some typos, such as Line 164, "0.51E-6 m2" should be '0.51 X 10-6m2' or '0.51 E-6 m2'.
A: Thank you. The sentence was corrected in the revised manuscript.

Reviewer 2 Report
1. Figure 1A and 1B - one figure is missing
2. The active surface of the optimized attachment delivers the desired forces while simultaneously extruding, rotating, or providing root control. Each attachment has a specific 78 surface profile on which the aligner provides pressure - how many sensors are used? Each type of force and different surfaces or in different cases - please add the details.
3. A port- 90 able microscope was be used to detect the area change....Please reliability and validity of the detection.
4. FEM figures of different varities of force and surfaces are needed.
5. Please separate results and discussions.
6. Please check this reference for relevant discussions: https://www.mdpi.com/2079-6412/12/10/1398
Author Response
Dear reviewer,
First, the authors would like to thank the time spent reviewing our manuscript entitled " Smart Flexible 3D sensor for monitoring orthodontics forces: Prototype Design and Proof of Principle experiment." We believe that all the considerations improved the quality of the manuscript, and we reiterate our wish to publish at Polymers. The answers to your comments are below point-by-point. The document was reviewed. The modifications were written in red in the revised manuscript.
REVIEWERS COMMENTS:
REVIEWER #2
- Figure 1A and 1B - one figure is missing
A: Thank you. The description for Figure 1 was corrected in the revised manuscript.
- The active surface of the optimized attachment delivers the desired forces while simultaneously extruding, rotating, or providing root control. Each attachment has a specific 78 surface profile on which the aligner provides pressure - how many sensors are used? Each type of force and different surfaces or in different cases - please add the details.
A: We appreciate your comment, but we hope the reviewer understands the proposed work as a proof of concept to measure orthodontic forces. The sensor dimension addressed in the paper (3 mm) is for demonstration purposes and can be reduced as long as the microscope resolution allows to detect its area change. The corresponding comment was added at the beginning of Section 2. As the reviewer pointed out, we expect that we need multiple miniaturized sensors to detect a general 3D force. Detailed design specifications will follow once we finalize the proof of concept. Thank you.
- A port- 90 able microscope was be used to detect the area change....Please reliability and validity of the detection.
A: Table 2 now presents error measures to indicate the difference from the FEM simulation. The average error is about 6%, with a standard deviation of about 13%. We understand this precision may not be sufficient for clinical needs, but as an early phase feasibility study, we believe it is a successful implementation using an easy concept of force measurement with plenty of improvement remain.
- FEM figures of different varities of force and surfaces are needed.
A: Please refer to the updated table 2, which presents error measures to indicate the difference from the FEM simulation. The deformed shape of the sensor in the FEM simulation was presented in Figure 2 already. We hope they suffice to answer this comment. Thank you.
- Please separate results and discussions.
A: Thank you for your suggestion.
- Please check this reference for relevant discussions: https://www.mdpi.com/2079-6412/12/10/1398
A: Thank you for your suggestion. The relevant reference was added to the revised manuscript.

Reviewer 3 Report
General comment:
This manuscript outlines an interesting work that is by using a 3D hemisphere as an visual-based force sensor. The reviewer has some concerns and found a few issues with the manuscript. The authors is advised to address the issue before the manuscript can be recommended for acceptance.
1. Figure 3 is of low quality. Suggest replacing with a higher resolution image and improving on the data presentation.
2. Similarly, the resolution of figure 4 is also very low. Suggest replacing wiht a higher resolution image.
3. Suggest adding a scale bar for figure 5.
4. For figure 6, what does the number on each scale mean? do they mean pixels? please add axis labels.
5. To the reviewer's knowledge, PDMS is usually thermoset polymer, however, the authors claimed that PDMS is thermoplastic polymer in line 85. Could it be a mistake? Otherwise, suggest clarifying this point in the manuscript.
6. It is not clear how the sensor can be useful in clinical application. This is because current technique requires a microscope and manual image processing to find out the contact area. Suggest adding a discussion to elaborate on how the workflow can be improved with appropriate technology.
7. Typically, 3D printed electronic sensors/transducer are used for sensing pressure. Suggest adding a discussion to compare how the proposed method is more advantageous compared to the other 3D printed soft sensors.
a. Singh, D., Tawk, C., Mutlu, R., Sariyildiz, E., Sencadas, V., & Alici, G. (2020, May). A 3d printed soft force sensor for soft haptics. In 2020 3rd IEEE International Conference on Soft Robotics (RoboSoft) (pp. 458-463). IEEE.
b. Zhao, W., Wang, Z., Zhang, J., Wang, X., Xu, Y., Ding, N., & Peng, Z. (2021). Vat photopolymerization 3D printing of advanced soft sensors and actuators: From architecture to function. Advanced Materials Technologies, 6(8), 2001218.
c. Shih, B., Mayeda, J., Huo, Z., Christianson, C., & Tolley, M. T. (2018, April). 3D printed resistive soft sensors. In 2018 IEEE International Conference on Soft Robotics (RoboSoft) (pp. 152-157). IEEE.
d. Guo Liang Goh, Wai Yee Yeong, Jannick Altherr, Jingyuan Tan, Domenico Campolo, 3D printing of soft sensors for soft gripper applications, Materials Today: Proceedings, 2022, ISSN 2214-7853, https://doi.org/10.1016/j.matpr.2022.09.025.
8. The authors should also discuss how they decide on the dimension of the hemisphere sensor.
Author Response
Dear Reviewer,
First, the authors would like to thank the time spent reviewing our manuscript entitled " Smart Flexible 3D sensor for monitoring orthodontics forces: Prototype Design and Proof of Principle experiment." We believe that all the considerations improved the quality of the manuscript, and we reiterate our wish to publish at Polymers. The answers to your comments are below point-by-point. The document was reviewed. The modifications were written in red in the revised manuscript.
REVIEWERS COMMENTS:
REVIEWER #3
General comment:
This manuscript outlines an interesting work that is by using a 3D hemisphere as an visual-based force sensor. The reviewer has some concerns and found a few issues with the manuscript. The authors is advised to address the issue before the manuscript can be recommended for acceptance
- Figure 3 is of low quality. Suggest replacing with a higher resolution image and improving on the data presentation.
A: Thank you. The figure resolution was increased.
- Similarly, the resolution of figure 4 is also very low. Suggest replacing wiht a higher resolution image.
- A: Thank you. The figure resolution was increased.
- Suggest adding a scale bar for figure 5.
A: Thank you. The figure was revised and the scale bar included.
- For figure 6, what does the number on each scale mean? do they mean pixels? please add axis labels.
A: Thank you. The figure was revised and the axis description included.
- To the reviewer's knowledge, PDMS is usually thermoset polymer, however, the authors claimed that PDMS is thermoplastic polymer in line 85. Could it be a mistake? Otherwise, suggest clarifying this point in the manuscript.
A: Thank you. The sentence as revised and corrected.
- It is not clear how the sensor can be useful in clinical application. This is because current technique requires a microscope and manual image processing to find out the contact area. Suggest adding a discussion to elaborate on how the workflow can be improved with appropriate technology.
A: Thank you. The discussion section was revised. Future efforts aim to address these challenges, integrate the corresponding electronic backbone for powering the sensor, signal processing, and wireless communication on a flexible wearable platform, and perform a large-scale orthodontic monitoring study. We will also investigate the sensor responses by unnormal force, possibly measured from the area change and sliding distance. Finally, a functional interface (handheld micro-scope device, database of captured images, and the force tracking record) will be devised to make the developed concept clinically applicable.
- Typically, 3D printed electronic sensors/transducer are used for sensing pressure. Suggest adding a discussion to compare how the proposed method is more advantageous compared to the other 3D printed soft sensors.
A: The proposed concept is electronics free and easy to implement, so it is advantageous and reliable, especially when it needs to be used in vivo and in a small scale. We added a comment at the beginning of Section 2 as: “This simple concept can operate more reliably compared to other flexible sensors and avoid complex implementation of resistive sensor or electronics in a small scale.”
- Singh, D., Tawk, C., Mutlu, R., Sariyildiz, E., Sencadas, V., & Alici, G. (2020, May). A 3d printed soft force sensor for soft haptics. In 2020 3rd IEEE International Conference on Soft Robotics (RoboSoft)(pp. 458-463). IEEE.
- Zhao, W., Wang, Z., Zhang, J., Wang, X., Xu, Y., Ding, N., & Peng, Z. (2021). Vat photopolymerization 3D printing of advanced soft sensors and actuators: From architecture to function. Advanced Materials Technologies, 6(8), 2001218.
- Shih, B., Mayeda, J., Huo, Z., Christianson, C., & Tolley, M. T. (2018, April). 3D printed resistive soft sensors. In 2018 IEEE International Conference on Soft Robotics (RoboSoft)(pp. 152-157). IEEE.
- Guo Liang Goh, Wai Yee Yeong, Jannick Altherr, Jingyuan Tan, Domenico Campolo, 3D printing of soft sensors for soft gripper applications, Materials Today: Proceedings, 2022, ISSN 2214-7853, https://doi.org/10.1016/j.matpr.2022.09.025.
- The authors should also discuss how they decide on the dimension of the hemisphere sensor.
A: Thank you. The discussion section was revised. The sensor dimension is for demonstration purposes and can be reduced as long as the microscope resolution allows for detecting its area change. This simple concept can operate more reliably compared to other flexible sensors and avoid complex implementation of resistive sensors or electronics on a small scale.

Round 2
Reviewer 2 Report
We believe that all the considerations improved the quality of the manuscript, and we reiterate our wish to publish at Polymers.......Polymers or Bioengineering?
Line 72 - still Figure 1 A is there. Something still missing.
Author Response
Dear reviewer # 2- 2nd round
First, the authors would like to thank the time spent reviewing our manuscript entitled " Smart Flexible 3D sensor for monitoring orthodontics forces: Prototype Design and Proof of Principle experiment." We believe that all the considerations improved the quality of the manuscript, and we reiterate our wish to publish at Bioengeneering. The answers to your comments are below point-by-point. The document was reviewed. The modifications were written in red in the revised manuscript.
REVIEWERS COMMENTS:
REVIEWER #2
We believe that all the considerations improved the quality of the manuscript, and we reiterate our wish to publish at Polymers.......Polymers or Bioengineering?
A: Thank you. We have corrected the cover letter.
Line 72 - still Figure 1 A is there. Something still missing.
A: Thank you . We have corrected the legend and call for the figure 1.
We very much appreciate your time and comments to our paper .
